# Solubility Determination of c-Met Inhibitor in Solvent Mixtures and Mathematical Modeling to Develop Nanosuspension Formulation

**DOI:** 10.3390/molecules26020390

**Published:** 2021-01-13

**Authors:** Maharjan Ravi, Tripathi Julu, Nam Ah Kim, Kyeung Eui Park, Seong Hoon Jeong

**Affiliations:** 1BK21 FOUR Team and Integrated Research Institute for Drug Development, College of Pharmacy, Dongguk University, Gyeonggi 10326, Korea; raavii@dgu.ac.kr (M.R.); julutripathi@gmail.com (T.J.); 2R&D Center, ABION Inc., Seoul 08394, Korea; pku1218@abionbio.com

**Keywords:** transcutol^®^ HP, thermodynamics, solubility, nanosuspension, mathematical models, precipitation

## Abstract

The solubility and dissolution thermodynamics of new c-Met inhibitor, ABN401, were determined in eleven solvents and Transcutol^®^ HP–water mixture (TWM) from 298.15 to 318.15 K. The experimental solubilities were validated using five mathematical models, namely modified Apelblat, van’t Hoff, Buchowski–Ksiazaczak *λh*, Yalkowsky, and Jouyban–Acree van’t Hoff models. The experimental results were correlated and utilized further to investigate the feasibility of nanosuspension formation using liquid anti-solvent precipitation. Thermodynamic solubility of ABN401 increased significantly with the increase in temperature and maximum solubility was obtained with Transcutol^®^ HP while low solubility in was obtained water. An activity coefficient study indicated that high molecular interaction was observed in ABN401–Transcutol^®^ HP (THP). The solubility increased proportionately as the mole fraction of Transcutol^®^ HP increased in TWM, which was also supported by a solvent effect study. The result suggested endothermic and entropy-driven dissolution. Based on the solubility, nanosuspension was designed with Transcutol^®^ HP as solvent, and water as anti-solvent. The mean particle size of nanosuspension decreased to 43.05 nm when the mole fraction of ABN401 in THP, and mole fraction of ABN401 in TWM mixture were decreased to 0.04 and 0.1. The ultrasonicated nanosuspension appeared to give comparatively higher dissolution than micronized nanosuspension and provide a candidate formulation for in vivo purposes.

## 1. Introduction

ABN401, (Figure 1, 4-[5-[4-[(4-Methylpiperazin-1-yl)methyl]phenyl]pyrimidin-2-yl]-2-[[5-(1-methylpyrazol-4-yl)triazolo[4,5-b]pyrazin-3-yl]methyl]morpholine, PubChem CID 118364782, C_29_H_34_N_12_O, molar mass 566.66 g·mol^−1^), is a next generation synthetic tyrosine kinase c-Met inhibitor, and showed its therapeutic potential in the treatment of non-small cell lung cancer by patient-derived xenograft model [1]. Unlike previous compounds of quinoline-containing chemical structures metabolized to form nephrotoxic poorly soluble metabolites, it is not degraded by aldehyde oxidase in human liver cytosol. However, this drug showed poor aqueous solubility, which may limit the drug release in gastrointestinal tract affecting drug absorption and bioavailability. Therefore, increasing solubility and dissolution rate for the drug could be a useful strategy to improve its bioavailability [2]. The solubility data of drugs in aqueous and organic solvents are crucial during preformulation studies and formulation development [3].

The solubility data of model drug ABN401 in any organic solvent or co-solvent mixture with respect to temperature were not available in the literature. However, previous study already reported that the drug was weakly basic compound with pKa and log P of 7.49 and 2.46, respectively [1]. Hence, in this study, the solubility of the model drug in methanol, ethanol, 1-propanol, 2-propanol, 1-butanol, 2-butanol, acetonitrile, acetone, ethyl acetate, Transcutol^®^ HP (THP), water, and in Transcutol^®^ HP–water mixture (TWM) was determined at temperatures ranging from 298.15 K to 318.15 K under atmospheric pressure using the static equilibrium method [3,4,5,6,7,8,9,10,11,12,13]. The modified Apelblat model (AM), van’t Hoff model (VHM), and Buchowski–Ksiazaczak *λh* model (BKM) were used to correlate the experimental solubility in selected organic solvents [14,15,16]. Similarly, for TWM, modified the AM, VHM, Jouyhan–Acree van’t Hoff model (JAVHM), and Yalkowsky model (YM) were also used to correlate the experimental mole fraction solubility [2,11]. Apparent thermodynamic properties including Gibbs free energy change ($\Delta G_{\mathrm{sol}}^{{}^{\circ}}$), enthalpy change ($\Delta H_{\mathrm{sol}}^{{}^{\circ}}$), and entropy change ($\Delta S_{\mathrm{sol}}^{{}^{\circ}}$) of the drug were calculated from the solubility data using VHM analysis for both organic solvents and binary mixed solvents [11,17,18].

Co-solvency or solvent mixing helps in estimating the preferential solvation of solute by the solvent compounds in mixtures [19,20,21,22,23,24,25,26]. Various co-solvents such as methanol, ethanol, polyethylene glycol (PEG) 400, acetone, ethyl acetate, dimethyl acetamide (DMA), dimethyl formamide (DMF), *N*-methyl-2-pyrrolidone (NMP), and dimethyl sulfoxide (DMSO) have been used to enhance the solubility of drugs [19,20,21,22,23,24,25,26,27]. Methanol, acetonitrile, DMA, DMF, and NMP fall under class 2 solvents while ethanol, 1-propanol, 2-propanol, 1-butanol, 2-butanol, acetone, and ethyl acetate fall under class 3 solvents [28]. Recently, THP has been extensively investigated as a co-solvent to enhance solubility of drugs in water co-solvent mixtures [3,9,10,11,12]. THP is a commonly used co-surfactant in the lipid-based formulations, such as the self-microemulsifying drug delivery system (SMEDDS), self-nanoemulsifying drug delivery system (SNEDDS), and nanosuspension [29,30,31,32,33]. Because of its low toxicity, enhanced solubilizing capacity, physiological compatibility, and being listed as excipient in the United States pharmacopoeia national formulary (USP NF), its application in pharmaceutical, cosmeceutical, and nutraceutical field is expanding [32,34]. It can be added as a co-solvent in the aqueous mixture to increase the solubility of drugs, which is very important in developing liquid based formulation [29,30,31]. In addition, for the model drug having a high melting point and high dose, nanosuspension was preferred over inclusion complex, and lipid-based systems such as SMEDDS, SNEDDS, solid lipid nanoparticle (SLN), and nanostructured lipid carrier (NLC) [35].

The objective of current study was to evaluate the solubility of ABN401, a model drug, in various solvents and solvent mixtures. It was investigated further based on the physicochemical properties using differential scanning calorimetry (DSC) and powder X-ray diffraction (PXRD). Based on the solubility data of ABN401 on various solvents, the least soluble (water) and the most soluble (THP) solvents were chosen to develop a stable nanosuspension using liquid anti-solvent precipitation [29,34,36]. It was a combination process of precipitation followed by microfluidization or ultrasonication. It resulted in nanocrystals, termed as nanosuspension, and their various properties including dissolution profile, particle size, and stability were evaluated.

## 2. Results and Discussion

### 2.1. Solid State Characterization

The DSC thermogram of the model drug is shown in Appendix A. The melting temperature (*T*_m_) of 413.09 ± 0.26 K and the enthalpy of fusion ($\Delta H_{\mathrm{fus}}$) of 20.32 ± 0.57 kJ·mol^−1^ appeared to agree with the previous studies [1]. Recovered solid solute from the bottom of the saturated solution also gave an endothermic peak, which was consistent with its initial form (Appendix A). Thermal properties of the initial and recovered drug were not significantly different (*p* > 0.05). As shown in Figure 2a, the initial PXRD pattern of the drug presented characteristic crystalline peaks at 7.94°, 10.40°, 12.37°, 13.86°, 15.96°, 18.78°, 19.89°, 20.59°, 21.01°, 24.82°, and 28.46° [37]. In the recovered solid solute from the bottom of saturated solution, the same diffraction peaks were observed, which appeared to suggest that there was no polymorphic transformation including solvate during the evaluation (Figure 2b–l, Appendix A).

### 2.2. Equilibrium Solubility

#### 2.2.1. Solubility in Organic Solvents

The experimental mole fraction solubility of the drug in organic solvents over the temperature range of 298.15–318.15 K is presented in Appendix A. For all solvents, the solubility appeared to increase with the increasing temperature (*p* < 0.05). Within the studied temperature range, the order of drug solubility was in the order of THP > acetone > 1-butanol > 1-propanol > 2-butanol > ethyl acetate > acetonitrile > 2-propanol > ethanol > methanol > water. THP appeared to show higher solubility, which was almost 1000 times greater than in water. It could be because of the low dielectric constant, low polarity, and higher molecular weight of THP compared to the other solvents [38]. THP has been used as a solvent in pharmaceuticals, cosmetics, and foods with low toxicity and strong solubilization effect [29]. However, polarity and dielectric constant are not the only factors responsible for increasing the solubility. Dissolution is a complex phenomenon that can be influenced by other factors including temperature, molecular structure of the drug and solvent, molecular size, solvent–solvent interaction, solute–solvent interaction, co-solvent ratio, and ability to form hydrogen bonding [33,36].

To understand the solvent effect on the drug solubility, a Kamlet–Taft linear solvation energy relationship (KAT-LSER) model with solvatochromatic parameters (α-hydrogen bond donor acidity, β-hydrogen bond acceptor basicity, and *π**-dipolarity or polarizability), and Hildebrand solubility parameter (*δ*_H_) was used in solvents as illustrated in Equation (1). The 2-propanol and 1-butanol appear to be statistically insignificant (*p* > 0.05). The solvatochromatic parameters for THP were not adequately reported in the literature, while the solubility in water was lower among the studied solvents. Hence, the solvents with statistically significant (*p* < 0.05) were only reported.
(1)$$ln\left(x_{e}\right)\;=\;c_{0}+c_{1}\alpha +c_{2}\beta +c_{3}\pi^{*}+c_{4}\left(\frac{V_{s}\delta_{H}^{2}}{100RT}\right)$$
where *c*_0_ is constant value, *c*_1_ and *c*_2_ are susceptibility of solute to solute–solvent interactions via hydrogen bonding, *c*_3_ and *c*_4_ are susceptibility of solute to electrostatic solute–solvent and solvent–solvent interactions, and *R*, *T*, and vs. are universal gas constants (8.314 J·K^−1^·mol^−1^), absolute temperature, and molar volume of solute, respectively. The vs. value for the drug was calculated as 26.5 *MPa*^1/2^ based on Fedors’ method (Appendix A) [39]. The parameters *α, β, π**, and *δ*_H_ were taken from published articles (Appendix A) [27,40,41]. The KAT-LSER model coefficient values with their standard error were estimated from multiple linear regression analysis of experimental and ideal mole fraction solubility data at 298.15 K.
(2)$$ln\left(x_{e}\right)\;=\;-16.32\left(1.11\right)-6.21\left(0.52\right)\alpha +11.66\left(0.99\right)\beta +6.18\left(0.91\right)\pi^{*}+11.25\left(5.90\right)\left(\frac{V_{s}\delta_{H}^{2}}{100RT}\right)$$
where as *n* = 14, *R*^2^ = 0.97, *F* = 91.51, and *RSS* = 0.34. Based on the estimated coefficients, the parameters *α, β, π**, and *δ*_H_ were 17.59%, 33.03%, 17.50%, and 31.86%, respectively. The *β, π**, and $\left(\frac{V_{s}\delta_{H}^{2}}{100RT}\right)$ indicated that hydrogen bonding interactions of solvent with solute, electrostatic solute–solvent interactions, and solvent–solvent interactions were all positive. The solute–solvent interactions and solvent–solvent interactions appeared to contribute more than non-specific electrostatic interactions. The negative *α* parameter appeared to indicate that increment in hydrogen bonding acidity of solvent decreased the solubility.

Experimental solubility data in each solvent were evaluated using different mathematical models such as AM, VHM, and BKM. Parameters of each model along with the relative mean standard deviation (*RMSD*) value are listed in Table 1 and the graphical representation of the calculated and experimental solubility of each model are described in Figure 3, Appendix A, and Appendix A. The smaller *RMSD* values in each model indicate a good agreement between the calculated and the experimental solubility; particularly, AM showed smaller *RMSD* value (0.171 × 10^−4^) than the other models.

#### 2.2.2. Solubility in Binary TWM Solvents

The values of mole fraction solubility of the drug in the TWM are provided in Appendix A. The maximum mole fraction solubility was observed at higher mole fraction of THP at 318.15 K (42.28 × 10^−4^), whereas the lowest solubility value was observed in water at 298.15 K (2.8 × 10^−6^). Figure 4 showed the trend of solubility increment with the increase in temperature and mole fraction of THP in the TWM (*p* < 0.05).

When w_2_ < 0.4, there was a slight increase in the solubility. Rapid rise was observed from w_2_ = 0.4 to w_2_ = 0.9. However, as the mole fraction of THP increases from 0.9 to 1, solubility slightly decreased. This appeared to indicate the importance of co-solvency to improve the solubility of the drug. Furthermore, the solubility of a solute in a mixed solvent was influenced by several factors such as polarity, temperature, mole fraction of solutes, and solvents [10].

Table 2 shows the parameters and mean relative deviation (*MRD*) (%) for AM, VHM, BKM, and JAVHM, and Table 3 shows the ln *x* values calculated by YM along with MRD. It was found that MRD (%) for AM (4.86%) was smaller compared to the other models and revealed a good agreement with the experimental data. Similarly, VHM and BKM also showed good fitting (5.03% and 5.80% *MRD*, respectively). However, these three models only considered the temperature, not the mole fraction of the co-solvent; therefore, these models were recommended only in the solvent, not in the mixed solvent. On the other hand, YM was used for calculating the solubility in mixed solvent systems. However, it may not be used to show temperature dependent solubility and showed high *MRD* value (>43%). Finally, JAVHM was chosen as the best model to calculate mole fraction solubility because it takes account of both the temperature and mole fraction of co-solvent.

### 2.3. Ideal Solubilities and Activity Coefficients

The activity coefficients (*γ*_i_) were calculated to study the molecular interactions between the drug and respective solvents. The *x*^idl^ values of the drug appeared to be significantly lower than *x*_e_ values in THP (*p* < 0.05). Meanwhile the *x*^idl^ values were appeared to be significantly higher than *x*_e_ values of the drug in water, methanol, ethanol, 1-propanol, 2-propanol, 1-butanol, 2-butanol, acetonitrile, acetone, and ethyl acetate (*p* < 0.05) (Table 4). At higher temperature, the *x*^idl^ values of the drug in 1-propanol, 1-butanol, 2-butanol, acetonitrile, acetone, and ethyl acetate appeared to be closer to *x*_e_ values of the drug (*p* > 0.05). Based on the observations, THP was selected for the solubility of the drug. The *γ*_i_ values of the drug were the lowest in THP. The *γ*_i_ values in the binary mixture of THP with water in various mole fraction were provided in Appendix A. The activity coefficient data supported the favorable solubility in the TWM mixture.

### 2.4. Apparent Thermodynamic Analysis

To evaluate the dissolution behavior of the drug in different solvents and the TWM binary mixture, thermodynamic analysis of solubility was performed [42]. In this study, Δ*H*°_sol_, $\Delta G_{\mathrm{sol}}^{{}^{\circ}}$, and $\Delta S_{\mathrm{sol}}^{{}^{\circ}}$ of the drug solution were obtained by VHM analysis with Equation (3) [26].
(3)$$\Delta H_{\mathrm{sol}}^{{}^{\circ}}=-R\left(\frac{\partial \ln x_{\exp}}{\partial \left(1/T-1/T_{\mathrm{hm}}\right)}\right)$$
where $x_{\exp}$ is the mole fraction solubility of the drug; R is the universal gas constant (8.314 J·mol^−1^·K^−1^); $T_{\mathrm{hm}}$ is the mean harmonic temperatures from 298.15 K to 318.15 K, and the value is 308.15 K. According to the VHM equation, the logarithm of mole fraction of the solute ($\ln x_{\exp}$) is linearly related to the reciprocal of the absolute temperature (1/*T*). The slope of the plot of $\ln x_{\exp}$ against $\left(1/T-1/T_{\mathrm{hm}}\right)$ gives the value of (−Δ*H*°_sol_/*T*) and the intercept helps in the calculation of $\Delta G_{\mathrm{sol}}^{{}^{\circ}}$ as expressed by the following equation.
(4)$$\Delta G_{\mathrm{sol}}^{{}^{\circ}}\;=-RT_{\mathrm{hm}}\;\times \mathrm{intercept}$$

Finally, the entropy change ($\Delta S_{\mathrm{sol}}^{{}^{\circ}}$) of drug dissolution can be obtained by the following equation:(5)$$\Delta S_{\mathrm{sol}}^{{}^{\circ}}\;=\left(\frac{\Delta H_{\mathrm{sol}}^{{}^{\circ}}-\;\Delta G_{\mathrm{sol}}^{{}^{\circ}}\;}{T_{\mathrm{hm}}}\right)$$

The positive values of $\Delta H_{\mathrm{sol}}^{{}^{\circ}}$ might suggest that the dissolution of the drug in the organic solvents was endothermic ($\Delta H_{\mathrm{sol}}^{{}^{\circ}}$ > 0) (Appendix A). In the solvents studied, mole fraction solubility of the drug increased with the increase in temperature. High values of $\Delta H_{\mathrm{sol}}^{{}^{\circ}}$ reflected the strong temperature-dependent solubility [43]. Moreover, positive $\Delta H_{\mathrm{sol}}^{{}^{\circ}}$ indicated that molecular interaction between the drug and solvents was stronger and required higher energies for breaking solute–solute and solvent–solvent intermolecular interaction [12]. Similarly, the decreased value of $\Delta G_{\mathrm{sol}}^{{}^{\circ}}$ indicates that the dissolution process is more favorable in the solvents with high solubility [25]. It was found that the $\Delta G_{\mathrm{sol}}^{{}^{\circ}}$ values were the highest in water and the lowest in THP, owing to the highest solubility of the drug in THP and the lowest solubility in water among the solvents. Dissolution of the drug showed the positive $\Delta S_{\mathrm{sol}}^{{}^{\circ}}$ value in methanol, ethanol, 1-propanol, 2-propanol, 2-butanol, and THP, whereas negative $\Delta S_{\mathrm{sol}}^{{}^{\circ}}$ values were obtained for 1-butanol, water, acetonitrile, acetone, and ethyl acetate. The positive $\Delta S_{\mathrm{sol}}^{{}^{\circ}}$ value of THP indicated entropy-driven dissolution while the negative $\Delta S_{\mathrm{sol}}^{{}^{\circ}}$ value of water indicated enthalpy-driven dissolution. This was further supported by Appendix A, where the mole fraction of THP in the TWM binary mixture produced the positive $\Delta S_{\mathrm{sol}}^{{}^{\circ}}$ value, which indicated entropy-driven dissolution of the drug [11].

The solvation behavior in various THP and water mixtures was evaluated using enthalpy–entropy compensation analysis (Figure 5). It was found that ABN401 in water, THP, and their various mixtures presented a positive slope where $\Delta H_{\mathrm{sol}}^{{}^{\circ}}$ values were directly proportional to $\Delta G_{\mathrm{sol}}^{{}^{\circ}}$ values. This might be because of the higher solvation of the drug in THP than the solvation behavior in water. The molecular interaction between the drug and THP was more dominant over interaction between the drug and water. The solvation behavior of the drug in the TWM mixture was consistent with the solvation behavior reported for other poorly soluble drugs [3,11,21,44].

The order of drug solubility in the selected solvents was the following: THP > acetone > 1-butanol > 1-propanol > 2-butanol > ethyl acetate > acetonitrile > 2-propanol > ethanol > methanol > water. It was supported by $\Delta G_{\mathrm{sol}}^{{}^{\circ}}$ values in Appendix A, which decreased as the solubility increased. Similar decrease in $\Delta G_{\mathrm{sol}}^{{}^{\circ}}$ values was observed in case of TWM binary mixture, where solubility increased as the molar fraction of THP gradually increased. Meanwhile the order of solvent polarity was in the following order: water > methanol > ethanol > THP > 1-propanol > 1-butanol > 2-propanol > 2-butanol > acetonitrile > acetone > ethyl acetate, and the solubility of the drug does not increase with increasing solvent polarity. It indicated that the dissolution was influenced not only by solvent polarity but also by interaction between solute–solvent molecules. The stearic hindrance of the alkyl group in the iso-alcohol (2-propanol, 2-butanol) molecules appeared to reduce drug solubility. The drug in TWM binary mixture had lower solubility than with THP solvent alone. The increase in drug solubility in THP may be because of the solubilizing effects of THP rather than solvent action. However, the TWM binary mixture had comparably superior solubilities than the other solvents considered in the study.

### 2.5. Inhibitory Effects of Polymer on Drug Precipitation

One major issue of nanosuspension is its change in concentration gradient of equilibrium solubility with time, leading to Ostwald ripening [45]. Such precipitation can be controlled by using polymer additives. The minimum solubility in water, maximum solubility in THP, and the decrease in solubility as the molar ratio of water in TWM binary mixture increased, gave useful information in formulating nanosuspension. The six different polymers/stabilizers were studied to inhibit drug precipitation while the dissolved drug in THP (solvent) was mixed with water (anti-solvent). Based on the previous studies, polymer screening, polymer ratio, solvent/anti-solvent ratio, and nanosuspension methods were selected [33].

The inhibitory effect of polymer/stabilizer was in the following order: hydroxypropyl β-cyclodextrin (HPβCD) > sodium lauryl sulfate (SLS) > Lutrol^®^ F127 > PEG 6000 > Kollidon^®^ K12 > Kollidon^®^ VA64. HPβCD and SLS appeared to give the maximum inhibitory effect on drug precipitation (Figure 6). Kollidon^®^ K12 and Kollidon^®^ VA64 were non-ionic polymers and are attached on the drug surface to occupy adsorption sites and prevent drug molecules from binding to crystal lattice in solution [46].

Hence, it appeared to act as a barrier to recrystallization. If the polymer concentration was inadequate, the adsorption sites might become exposed to solution. Thus, crystal growth could occur, and aggregation could take place. On the contrary, if the polymer concentration was in excess, drug surfaces would become thicker, shielding from the solution, and thus diffusion between solvent and anti-solvent might be suppressed [29]. This would increase the attraction between colloidal particles and lead to particle growth. Therefore, surfactant was included to reduce the surface tension in solid–liquid interface. It appeared to increase the nucleation rate and reduce the particle size. The surfactant appeared to reduce the hydrophobic interaction, making the drug less hydrophobic. SLS, an anionic surfactant, appeared to increase the repulsive force between the particles to increase the barrier, preventing particle growth and aggregation [47].

### 2.6. Formation of Nanosuspension by Liquid Anti-Solvent Precipitation

The size and morphology of the drug molecule and its formulated nanosuspension were illustrated in Figure 7. The supplied drug molecule appeared to have 300 μm average particle size. It was formulated into nanosuspension. The lower mole fraction of the drug in the THP (*X*_1_) and in TWM mixtures (*X*_2_) appeared to give nanosuspension with smaller mean particle size. The drug solubility increased gradually when the mole fraction of THP in the TWM mixture was > 0.2. To efficiently formulate nanosuspension by liquid anti-solvent precipitation, the ratio of solvent to anti-solvent should be <0.2 [27]. When *X*_1_ = 0.04 and *X*_2_ = 0.1, the prepared nanosuspension had 43.05 nm mean particle size. The mole fraction of THP in TWM mixture at 0.1 (*X*_2_ = 0.1) appeared to have the lowest solubility (Appendix A), and thus, resulted in smaller mean particle size. The experimental results appeared to be consistent with the previous studies [27].

The characterization of zeta potential and in vitro dissolution are illustrated in Appendix A, Table 5, and Figure 7. The particle size, polydispersity index (PDI), and zeta potential of nanosuspensions (F1 to F4) appeared to be in the range of 43.05 to 120.10 nm, 0.29 to 0.34, and −34.57 to −43.07 mV, which suggested that such formed nanosuspensions were stable. The ultrasonicated F1, F2, and F4 formulations appeared to give >97% dissolution rate while the microfluidized formulations appeared to give >92% (*p* < 0.05). The ultrasonicated F2 formulation appeared to give 87.69% release within 15 min while microfluidized F2 formulation appeared to give 84.27% release within 15 min. The ultrasonicated nanosuspension appeared to give comparatively higher dissolution than microfluidized nanosuspension.

## 3. Experimental Section

### 3.1. Materials

ABN401 was kindly supplied from Abion Inc. (Seoul, Korea). THP was obtained from Gattefosse (Cedex, France). Methanol, ethanol, and acetonitrile were obtained from Avantor Performance Materials (Center Valley, PA, USA). 1-Propanol, 2-propanol, 2-butanol, acetone, and SLS were purchased from Daejung Chemical & Metals Co., Ltd. (Siheung, Korea). 1-Butanol and ethyl acetate were purchased from Junsei Chemical Co., Ltd. (Tokyo, Japan). Detailed information of ABN401 and solvents is provided in Appendix A. PEG 6000 and HPβCD were purchased from Sigma-Aldrich (St. Louis, MO, USA). Lutrol^®^ F127, Kollidon^®^ VA64, and Kollidon^®^ K12 were purchased from BASF (Ludwigshafen, Germany). The water was collected from a Milli-Q water purifier (Millipore, Lyon, France). All reagents were of analytical or high-performance liquid chromatography (HPLC) grade and were used as received.

### 3.2. High Performance Liquid Chromatography

Purity of ABN401 was tested using an HPLC system (LC-20AD, Shimadzu, Kyoto, Japan) with Eclipse plus C_18_ column (4.6 mm × 150 mm, 5 µm) set at a temperature of 30 °C and the ultraviolet (UV) detector at 282 nm. The mobile phase was a mixture of acetonitrile and 50 mM acetate buffer at pH 5.0 (50:50% *v*/*v*). The flow rate of the mobile phase was 0.5 mL·min^−1^ and the injection volume was 10 µL. All measurements were performed in triplicate.

### 3.3. Solid State Characterization

Melting temperature and enthalpy of fusion for samples were determined using differential scanning calorimetry (DSC) (TA Instruments, New Castle, DE, USA). For the DSC analysis, the sample (2 mg) was accurately weighed (Mettler Toledo, Greifensee, Switzerland) and sealed in a Tzero Aluminum Pan. A blank pan was employed as a reference. DSC measurements were carried out at a scan rate of 10 K·min^−1^ from 293.15 K to 453.15 K under a nitrogen flow of 50 mL·min^−1^. The standard uncertainty of melting temperature was estimated to be 0.5 K. Various thermal parameters were obtained and interpreted using the software provided with the instrument. The thermal analysis was performed to analyze different thermal parameters and to evaluate the possible transformations of ABN401 into its polymorph/solvate/hydrate. ABN401 solid solute was recovered from the bottom of saturated solution by slow evaporation of the solvent at 298.15 K [11,42,48].

Powder X-ray diffraction (PXRD) patterns were measured using a D2 phaser benchtop X-ray diffractometer (Bruker AXS GmbH, Karlsruhe, Germany) equipped with a Ni-filtered Cu-Kα radiation (λ = 1.54056 Å) and a high speed LynxEye detector. The powder samples were placed in a quartz holder and scanned over a range of 4–40° at a scanning rate of 6°/min.

### 3.4. Solubility in Different Organic Solvents

The solubility of ABN401 in various solvents (water, methanol, ethanol, 1-propanol, 2-propanol, 1-butanol, 2-butanol, acetonitrile, acetone, ethyl acetate, THP) and in the TWM binary mixture was conducted using static equilibrium method at different temperature ranges from 298.15 to 318.15 K [49]. The experimental conditions and the procedures were based on the previously published articles [44,50]. Briefly, the model drug was added in an excess amount in 5 mL glass vial containing 2 mL of the solvent. Each vial was tightly closed and sealed with parafilm. The solid–solvent mixtures were vortexed for 10 min, using a vortex shaker (Daihan Scientific, Seoul, Korea). It was followed by incubation in a shaking water bath (Jeiotech Co., Ltd., Daejeon, Korea) at 100 rpm for 72 h to reach equilibrium. The water bath was provided with a thermostat (Shanghai Laboratory Instrument Works, Shanghai, China) capable of maintaining temperature within ±0.05 K. The samples were kept stable to allow undissolved particles to settle down at the bottom. The experiment was carried out in triplicate and arithmetic average was used as the final value. It was then centrifuged at 10,000 rpm for 10 min (Eppendorf Inc., Westbury, CT, USA). Supernatants were then filtered through a 0.45-μm polytetrafluoroethylene (PTFE) syringe filter (Hyundai Micro, Seoul, Korea) and appropriately diluted with respective solvent before analysis.

Quantification of the drug was carried out with a previously validated HPLC method [1]. The standard calibration curve was found to be linear in the range of 1.6 μg·mL^−1^ to 50 μg·mL^−1^ with a correlation coefficient of 0.9999.

All measurements were performed in triplicate where the average values were used to calculate mole fraction solubility of the drug. The experimental mole fraction solubility ($x_{\exp}$) of the drug in organic solvents was calculated using Equation (6) [3]:(6)$$x_{\exp}=\;\frac{m_{A}/M_{A}}{m_{A}/M_{A}\;+\;m_{1}/M_{1}}$$
where $m_{A}$ and $m_{1}$ are the mass of the drug and solvent, $M_{A}$ and $M_{1}$ are the respective molar mass of the drug and solvent, respectively.

The mole fraction of THP ($w_{2}$) in the binary solvents varied from 0.1 to 0.9 and it can be obtained by Equation (7) [3]:(7)$$w_{2}=\;\frac{m_{2}}{m_{2}+\;m_{1}}$$
where $m_{1}$ and $m_{2}$ represent the mass of water and THP, respectively. Similarly, the mole fraction solubility of the drug ($x_{\exp}$) in the binary mixture of water and THP at different temperatures can be obtained by Equation (8) [3]:(8)$$x_{\exp}=\;\frac{m_{A}/M_{A}}{m_{A}/M_{A}\;+m_{1}/M_{1}+m_{2}/M_{2}}$$
where $m_{A}$, $m_{1}$, and $m_{2}$ are the mass of the drug, water, and THP; $M_{A}$, $M_{1}$, and $M_{2}$ are the molar mass of the drug, water, and THP. The experiment was carried out in triplicate and arithmetic average was used as the final value.

### 3.5. Ideal Solubilities and Activity Coefficients

The *x*^idl^ value of the drug was calculated using Equation (9).
(9)$$ln\;x^{\mathrm{idl}}=\frac{-\Delta H_{\mathrm{fus}}\left(T_{\mathrm{fus}}+T\right)}{RT_{\mathrm{fus}}T}+\left(\frac{\Delta C_{p}}{R}\right)\left[\frac{T_{\mathrm{fus}}-T}{T}+ln\left(\frac{T}{T_{\mathrm{fus}}}\right)\right]$$
where, *R* = universal gas constant and the other parameters were explained in previous articles [51,52]. The $\Delta C_{p}$ of the drug was calculated with Equation (10).
(10)$$\Delta C_{p}=\frac{\Delta H_{\mathrm{fus}}}{T_{\mathrm{fus}}}$$

The *T*_fus_ and $\Delta H_{\mathrm{fus}}$ values for the drug were calculated as 413.09 K and 20.32 kJ·mol^−1^, respectively, using DSC analysis. The $\Delta C_{p}$ of the drug was obtained as 49.19 J·mol^−1^K^−1^. The *x*^idl^ values of the drug could be calculated using Equation (9) and the $\gamma_{i}$ values in different solvents were calculated using Equation (11) [51].
(11)$$\gamma_{i}=\frac{x^{\mathrm{idl}}}{x_{e}}$$

### 3.6. Thermodynamic Models

The solubility of ABN401 in organic solvents was analyzed and correlated using modified AM, VHM, and BKM, and solubility of ABN401 in THP mixtures was correlated using modified AM, VHM, BKM, JAVHM, and YM.

#### 3.6.1. Modified Apelblat Model

Modified AM is a semi-empirical model. Equation (12) correlates mole fraction solubility and the absolute temperature for both the polar and non-polar solvents. It can be expressed as [14,15,17,38].
(12)$$\ln x_{1}=A+\;\frac{B}{T}+C\;\ln\left(T\right)$$
where $x_{1}$ is the mole fraction solubility of the drug at absolute temperature *T* (K), and *A*, *B*, and *C* are the model parameters obtained by non-linear regression analysis. The parameters *A* and *B* represent the non-ideal behavior of the solution in terms of variation of activity coefficient in the solution, and *C* reflects the effect of temperature on the enthalpy of fusion.

#### 3.6.2. Van’t Hoff Model

In the VHM equation illustrated as in Equation (13), logarithm of mole fraction solubility of the solute is linearly correlated to the reciprocal of the absolute temperature in the ideal solution. It is a simplified expression of activity coefficient formula and expressed as [18]:(13)$$\ln x_{1}=\;a+\frac{b}{T}$$
where *T* is the absolute temperature, $x_{1}$ is mole fraction solubility of ABN401, and *a* and *b* are the model parameters.

#### 3.6.3. Buchowski–Ksiazaczak λh Model

To describe the solid–liquid equilibrium behavior of the solute, BKM was developed and Equation (14) was obtained by Buchowski. The equation is as following [16]:(14)$$\ln\left[1+\frac{\lambda \left(1-x_{1}\right)}{x_{1}}\right]=\lambda h\left[\frac{1}{T}-\;\frac{1}{T_{m}}\right]$$
where $x_{1}$ is the mole fraction solubility of the drug, T is the experimental absolute temperature, and $T_{m}$ is the melting temperature (Kelvin) of the drug. The value of $T_{m}$ was found to be 413.09 K with the thermal analysis. The parameters λ and h are the model parameters.

#### 3.6.4. Yalkowsky Model

Experimental mole fraction solubility in the mixed solvents can be calculated by YM by using Equation (15). The equation is given as [2]:(15)$$\ln x_{m}=w_{1}\ln x_{1}+w_{2}\ln x_{2}$$
where $x_{1}$ and $x_{2}$ are the mole fraction solubility of ABN401 in water and THP; $x_{m}$ is the mole fraction solubility of the drug in binary solvent mixtures; $w_{1}$ and $w_{2}$ are the mole fractions of water and THP without the drug.

#### 3.6.5. Jouyban–Acree Van’t Hoff Model

The JAVHM equation is the combination of the JAM equation and the VHM equation. This combined equation is widely used to describe the relationship between the mole fraction solubility and temperature composition of the solute in the mixed solvents. The basic JAM equation to determine the drug solubility in binary mixed solvents at different temperature is given as [22]:(16)$$\ln x_{m,T}=\;w_{1}\ln x_{1,T}+w_{2}\ln x_{2,T}+\frac{w_{1}w_{2}}{T}\overset{n}{\underset{i=0}{\sum }}Ji\left(w_{1}-w_{2}\right)x^{i}$$
where $x_{m,T}$
$x_{1,T}$
$x_{2,T}$ are the mole fraction solubility of ABN401 in binary solvent mixtures, water, and THP at temperature *T* and *J*_i_ is the model constant calculated by multiple linear regression of $\ln x_{m,T}-\;w_{1}\ln x_{1,T}-w_{2}\ln x_{2,T}\;\mathrm{vs}.\;\frac{w_{1}w_{2}}{T}$, $\frac{\left(w_{1}w_{2}\left(w_{1}-w_{2}\right)\right)}{T}$, and $\frac{\left(w_{1}w_{2}{\left(w_{1}-w_{2}\right)}^{2}\right)}{T}$.

On combining Equation (16) with the van’t Hoff model, a new equation can be obtained as [23,24]:(17)$$\ln x_{\exp,T}=\;\propto_{1}w_{1}+\frac{\propto_{2}w_{1}}{T}+\propto_{3}w_{2}+\frac{\propto_{4}w_{2}}{T}+\frac{J_{0}\left(w_{1}-w_{2}\right)}{T}+\frac{J_{1}\left(w_{1}w_{2}\left(w_{1}-w_{2}\right)\right)}{T}+\frac{J_{2}(w_{1}w_{2}{\left(w_{1}-w_{2}\right)}^{2})}{T}$$
where $\propto_{1}$, $\propto_{2}$, $\propto_{3}$, $\propto_{4}$, $J_{1}$, $J_{2}$, and $J_{3}$ are the model parameters.

#### 3.6.6. Data Correlation

In order to distinguish the experimental and calculated solubility data, *RMSD* was used, which is expressed as [23,24]:(18)$$MRD\;\left(\%\right)=\;\frac{100}{N}\;\sum (\frac{\left[x_{\exp}-x_{\mathrm{cal}}\right]}{x_{\exp}})$$
(19)$$RMSD=\sqrt{\frac{\sum_{i=1}^{N}{\left(x_{\exp}-\;x_{\mathrm{cal}}\right)}^{2}}{N}}$$
where *N* is number of experimental data points, and $x_{\exp}$ and $x_{\mathrm{cal}}$ represent experimental value and calculated values of mole fraction solubility of the drug, respectively.

### 3.7. Inhibitory Effect of Polymer on Drug Precipitation

The inhibitory effect of polymers on the precipitation of ABN401 was measured using the USP dissolution apparatus 2 (paddle) at 100 rpm using 500 mL of distilled water containing polymers at 0.5% *w*/*v* maintained at 37 ± 0.5 °C (Agilent Technologies, Santa Clara, CA, USA). Kollidon^®^ VA64, Kollidon^®^ K12, Lutrol^®^ F127, HPβCD, PEG 6000, and SLS were selected as polymers/stabilizers [29,33,53]. The experimental conditions were the same as is mentioned in Section 2.5. The samples were filtered using a 0.45-μm PTFE syringe filter, diluted with methanol, and analyzed using the HPLC system.

### 3.8. Preparation of Nanosuspension

The nanosuspension was prepared using the liquid anti-solvent precipitation method [29,30,31,36]. The concentration and polymer ratio were selected from the previously reported study [29]. The mole fractions of 0.04 and 0.08 drug concentration in THP were prepared separately as illustrated in Appendix A [29]. An aqueous solution was also prepared by dispersing Kollidon^®^ VA and Kollidon^®^ K12 with individual polymers like Lutrol^®^ F127, HPβCD, and PEG 6000 in 1:0.5:1 ratio, or stabilizer like SLS in 1:0.5:0.1 ratio as mentioned in Appendix A [29]. The screening study, formulation, and process conditions were selected based on the previous studies [12,34]. The drug–THP solution was added dropwise at a rate of 1 mL·min^−1^ to the polymer/stabilizer aqueous solution, with magnetic stirring. The two samples were prepared for each drug concentration at solvent/anti-solvent ratios of 1:4 and 1:9. It was stirred for 1 h. The prepared suspension was divided into two halves. One part was ultrasonicated using an ultrasonicator at 200 W for 30 min under ice bath (Sonics & Materials Inc, Newtown, CT, USA). The other half was microfluidized using a microfluidizer at 20,000 psi for 20 cycles, under ice bath (Microfluidics, Westwood, MA, USA) [54]. The procedure is illustrated in Figure 7.

### 3.9. Dynamic Light Scattering

The particle size and PDI were measured using a Zetasizer dynamic light scattering (DLS) instrument (Malvern Instruments Ltd., Worchestershire, UK), equipped with He-Ne laser at 633 nm at a scattering angle of 90°. DLS can be useful to determine the particle size of nanoparticles, their distribution in suspension, and zeta potential at the surface of nanoparticles. The nanosuspension was diluted 500 times and allowed to be stabilized for 30 min. Analysis was performed in triplicate for each sample (30 runs in each measurement) and the values were provided as a mean of triplicate samples. Zeta potential was determined using the laser Doppler method to evaluate physical stability of colloidal systems.

### 3.10. In Vitro Dissolution Study

The In vitro dissolution test was performed in 500 mL of simulated gastric fluid (pH 1.2) with paddle apparatus at 37 ± 0.5 °C and 100 rpm (Agilent Technologies, Santa Clara, CA, USA). The 10-mL nanosuspension was added into the dissolution vessels (n = 6), and the samples were withdrawn at predetermined time intervals. The equivalent amount of aliquot was replaced with fresh medium in the dissolution vessel each time. The sink condition was maintained throughout the experiment. The aliquots were filtered through a 0.45-µm PTFE syringe filter. The samples were analyzed using the HPLC system. All readings were the mean and standard deviation of six samples.

### 3.11. Scanning Electron Microscope

The nanosuspension was freeze-dried using 5% (*w*/*v*) lactose as a cryoprotectant in a freeze dryer (Operon, Yangchon, Korea) for 72 h [55]. The morphology of dried powder was examined with a scanning electron microscope (SEM) instrument (COXEM, Daejeon, Korea) at an accelerating voltage of 20 kV. The samples were initially coated with gold under vacuum in an argon atmosphere before the examination.

## 4. Conclusions

The solubility of the drug was determined in eleven solvents and in TWM mixture using a static equilibrium method and correlated with various models, and modified Apelblat model showed good agreement. The solubility of the drug increased with an increase in temperature for all solvents including the TWM mixture. Based on the KAT-LSER model, the drug solubility decreased as the hydrogen bond acidity (*α*) of the solvent increased. The activity coefficients indicated that THP–drug had the maximum number of interactions and, thus, THP was the best solvent. Thermodynamic analysis suggested endothermic and entropy-based dissolution. Based on the solubility, THP and water were used as solvent and anti-solvent to prepare the nanosuspension using liquid anti-solvent precipitation. The mean particle size of the nanosuspension could be controlled by adjusting the mole fraction of the drug in THP, and mole fraction of the drug in the TWM mixture. The ultrasonicated nanosuspension appeared to give a comparatively higher dissolution rate than micronized one. The solubility data and observations could be useful for particle size control, purification, crystallization, and new formulation development for further studies.

## Figures and Tables

**Figure 1 molecules-26-00390-f001:**
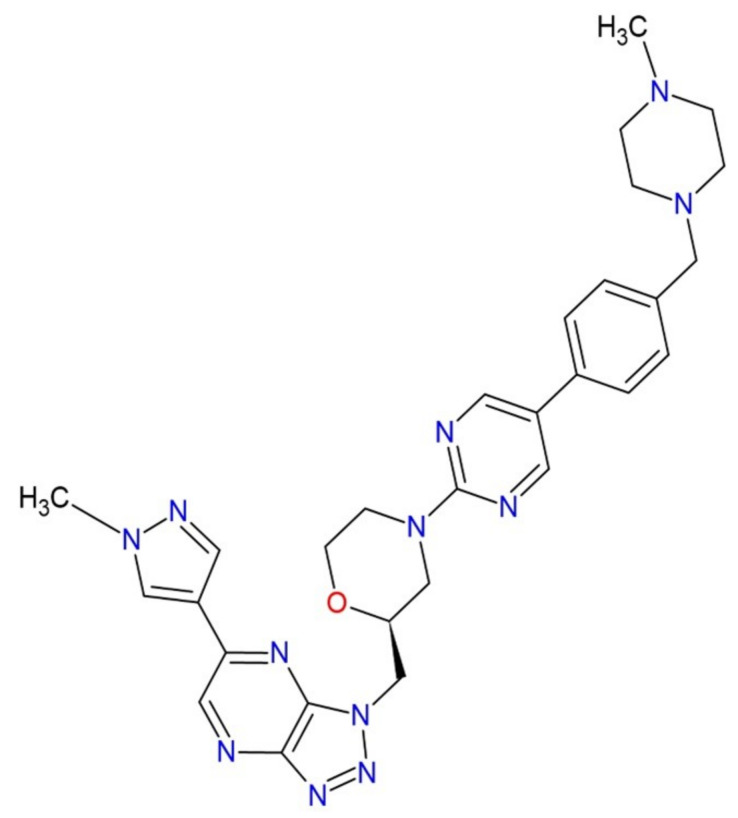
Chemical structure of ABN401.

**Figure 2 molecules-26-00390-f002:**
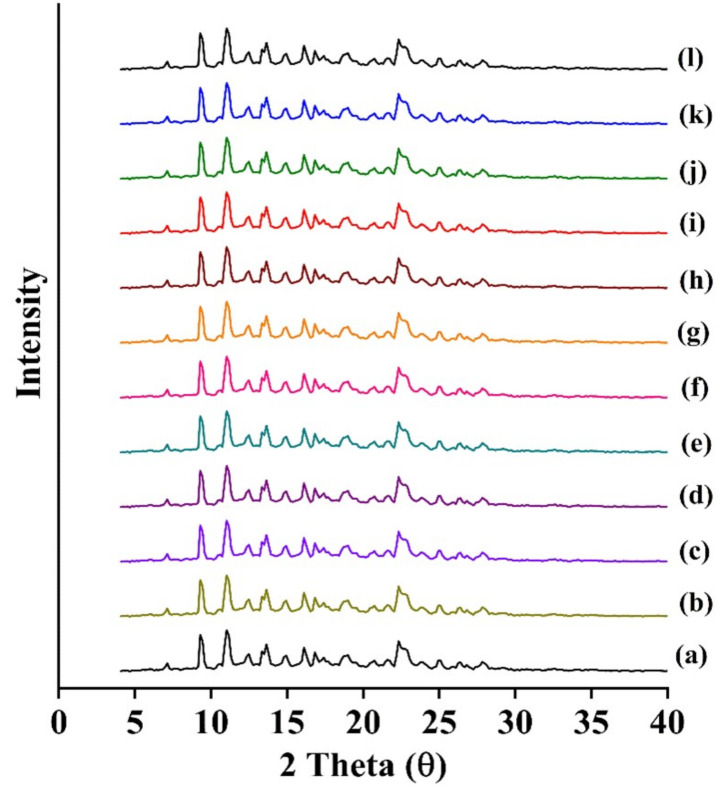
Powder X-ray diffraction (PXRD) patterns of the model drug alone (**a**), the drug—recovered from water (**b**), methanol (**c**), ethanol (**d**), 1-propanol (**e**), 2-propanol (**f**), 1-butanol (**g**), 2-butanol (**h**), acetonitrile (**i**), acetone (**j**), ethyl acetate (**k**), and Transcutol^®^ HP (THP) (**l**).

**Figure 3 molecules-26-00390-f003:**
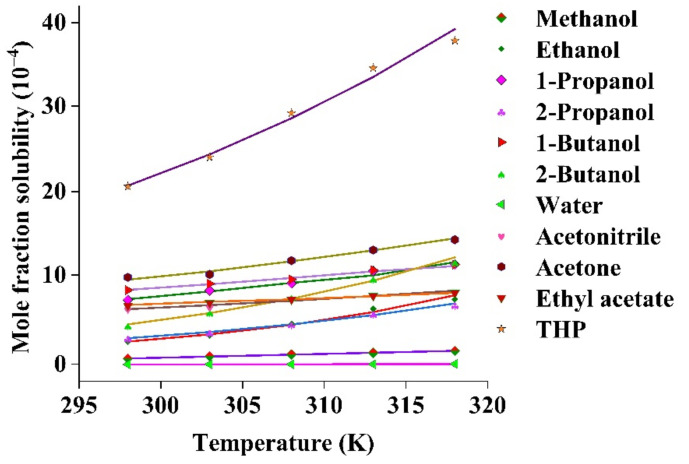
Experimental and calculated mole fraction solubility of the drug in organic solvents based on BKM. Solid lines denote the calculated solubility.

**Figure 4 molecules-26-00390-f004:**
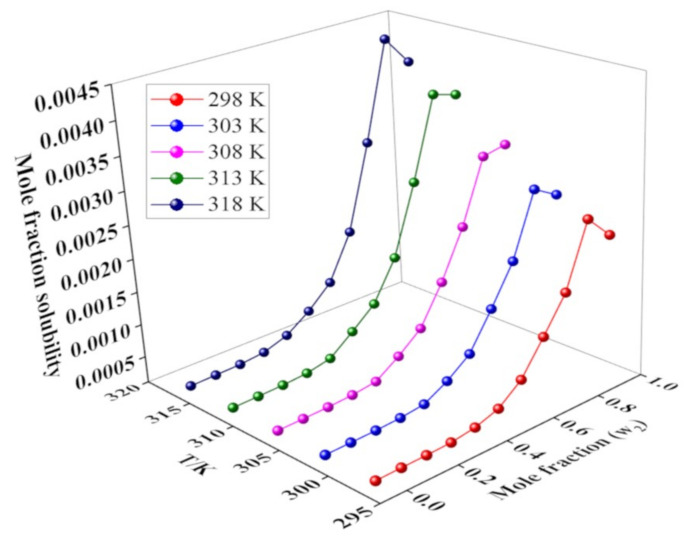
Impact of mole fraction of THP (*m*) on the mole fraction solubility of the drug at different temperatures.

**Figure 5 molecules-26-00390-f005:**
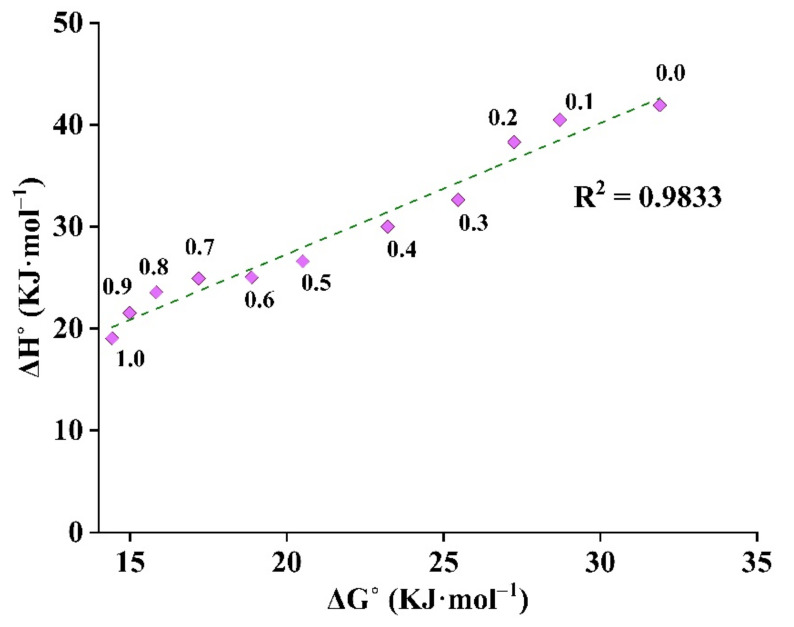
Enthalpy–entropy compensation analysis in different mole fractions of THP in the TWM binary mixture at *T*_hm_ of 308.15 K. The mole fraction of THP in TWM mixture was represented from 0.0 to 1.0.

**Figure 6 molecules-26-00390-f006:**
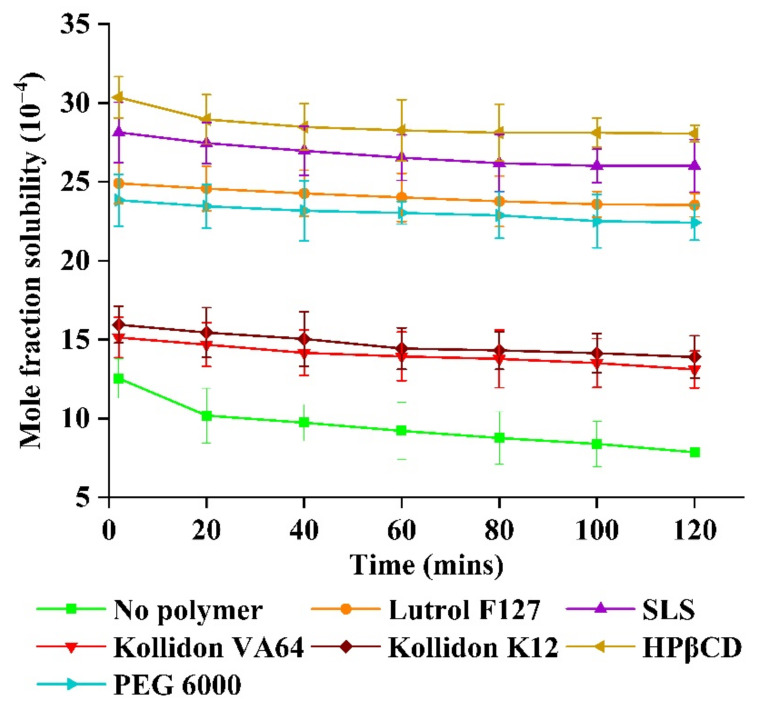
Inhibitory effects of various polymers and stabilizers on drug precipitation.

**Figure 7 molecules-26-00390-f007:**
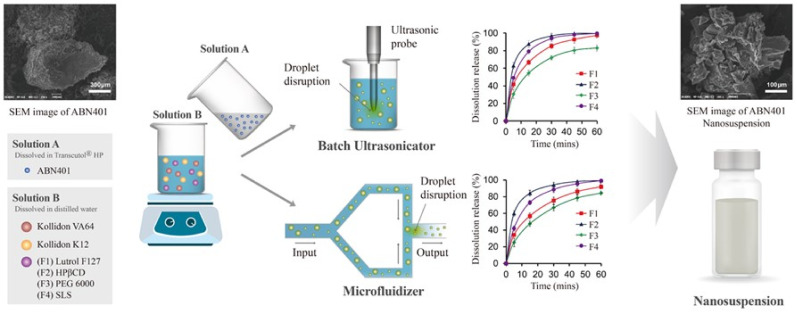
Schematic diagram of nanosuspension formation using liquid anti-solvent precipitation.

**Table 1 molecules-26-00390-t001:** Parameters of the modified Apelblat model (AM) equation, van’t Hoff model (VHM) equation and Buchowski–Ksiazaczak *λ**h* model (BKM) equation for ABN401 in organic solvents and their respective relative mean standard deviation (RMSD) values.

Solvents	AM				VHM			BKM		
	*A*	*B*	*C*	RMSD*10^−4^	*a*	*b*	RMSD*10^−4^	*λ* *10^−2^	*h**10^−3^	RMSD*10^−4^
Water	−374.54	14,006.70	55.25	0.001	−2.72	−2997.05	0.001	0.003	81,500	0.001
Methanol	490.45	−26,827.80	−72.07	0.032	5.46	−4646.29	0.013	0.294	1552.96	0.014
Ethanol	787.98	−41,435.30	−115.39	0.189	11.44	−5922.71	0.276	5.973	99.60	0.291
1-Propanol	273.94	−14,746.60	−40.67	0.132	0.25	−2229.99	0.163	0.298	581.15	0.187
2-Propanol	205.50	−13,595.70	−29.52	0.086	6.88	−4512.17	0.102	1.643	268.80	0.108
1-Butanol	−65.44	1399.93	9.42	0.224	−2.07	−1497.91	0.108	0.061	954.68	0.111
2-Butanol	841.86	−43,078.50	−123.77	0.198	8.97	−4988.55	0.250	4.456	110.99	0.270
Acetonitrile	120.15	−7033.34	−18.25	0.138	−2.66	−1417.25	0.140	0.040	1349	0.157
Acetone	−318.55	12,641.10	47.25	0.230	−0.58	−1900.52	0.241	0.225	570.21	0.233
Ethyl acetate	−31.99	264.30	4.17	0.029	−3.93	−1018.83	0.017	0.019	1936.12	0.026
THP	456.50	−23,707.90	−67.25	0.620	3.92	−3010.37	0.777	2.587	106.47	0.859
Overall	0.171				0.190			0.205		

* Relative uncertainties, *u(A)* = 3.04, *u(B)* = 4.95, *u(C)* = 3.07, *u(a)* = 0.13, *u(b)* = 0.19, *u(λ)* = 0.02, *u(h)* = 3.19.

**Table 2 molecules-26-00390-t002:** Parameters of the modified AM equation, VHM equation, BKM equation and Jouyhan–Acree van’t Hoff model (JAVHM) equation in the Transcutol^®^ HP–water mixture (TWM) mixture.

*w* _2_	AM			VHM		BKM	
	*A*	*B*	*C*	*a*	*b*	*λ**10^−3^	*h**10^−3^
0	−374.974	14,025.4	55.2935	−2.84474	−2999.98	0.018	117,563
0.1	−210.273	7229.53	30.6005	−4.32968	−2192.64	0.021	55,381.8
0.2	1291.7	−62,115.9	−192.138	−1.40214	−2954.84	0.135	17,284.6
0.3	845.029	−42,649.8	−125.119	2.97107	−4124.51	0.702	4819.56
0.4	−3387.62	150,906	503.981	4.20757	−4273.88	2.208	1672.14
0.5	1816.02	−86,808.6	−269.254	3.92576	−3902.93	4.872	693.367
0.6	339.704	−17,966	−50.5433	−0.456256	−2403.26	1.494	1003.25
0.7	57.707	−4651.25	−8.79499	−1.483921	−1943.19	1.193	706.422
0.8	−615.06	25,244.2	91.7279	−2.27488	−3019.63	9.504	223.656
0.9	−1264.99	55,224.9	188.166	1.38631	−2713.24	10.772	169.065
1	456.071	−23,804.5	−67.4966	1.81408	−3021.68	15.432	145.412
MRD (%)	4.869			5.032		5.804	
JAVHM							
Parameters	*α* _1_	*α* _2_	*α* _3_	*α* _4_	*J* _1_	*J* _2_	*J* _3_
Value	−0.97	−3128.83	4.12	−2907.49	86.86	−1664.92	−1593.51
MRD (%)	7.08						

* Relative uncertainties, *u(A)* = 2.95, *u(B)* = 3.17, *u(C)* = 3.31, *u(a)* = 0.16, *u(b)* = 0.09, *u(λ)* = 0.03, *u(h)* = 4.06.

**Table 3 molecules-26-00390-t003:** Ln *x* values of the drug calculated by the Yalkowsky model (YM) equation in the THP mixture at different temperatures.

*w* _2_	Ln *x*				
	298.15 K	303.15 K	308.15 K	313.15 K	318.15 K
0	−12.78	−12.61	−12.46	−12.34	−12.12
0.1	−12.12	−11.95	−11.80	−11.67	−11.47
0.2	−11.46	−11.29	−11.14	−11.00	−10.81
0.3	−10.80	−10.63	−10.47	−10.33	−10.16
0.4	−10.14	−9.98	−9.81	−9.67	−9.50
0.5	−9.48	−9.32	−9.15	−9.00	−8.85
0.6	−8.82	−8.66	−8.48	−8.33	−8.19
0.7	−8.16	−8.00	−7.82	−7.66	−7.54
0.8	−7.50	−7.34	−7.16	−7.00	−6.88
0.9	−6.84	−6.69	−6.50	−6.33	−6.23
1	−6.19	−6.03	−5.83	−5.66	−5.58
MRD (%)	43.43	43.69	43.10	44.37	44.12
Overall	43.75				

* Standard uncertainties, *u(T)* = 0.04 K.

**Table 4 molecules-26-00390-t004:** Activity coefficients (*γ*_i_) of the model drug in various solvents at 298.15 to 318.15 K.

Solvents	*γ* _i_				
	*T* = 298.15 K	*T* = 303.15 K	*T* = 308.15 K	*T* = 313.15 K	*T* = 318.15 K
Water	887.91	609.19	431.99	314.50	211.50
Methanol	65.00	38.30	24.84	16.74	10.91
Ethanol	11.55	6.99	3.90	2.28	1.60
1-Propanol	3.53	2.48	1.84	1.31	1.02
2-Propanol	9.72	6.29	3.96	2.52	1.76
1-Butanol	3.03	2.26	1.75	1.30	1.02
2-Butanol	6.14	3.61	2.25	1.43	0.99
Acetonitrile	4.12	3.23	2.32	1.80	1.43
Acetone	2.56	2.01	1.42	1.06	0.81
Ethyl acetate	3.90	3.01	2.34	1.82	1.45
THP	1.22	0.85	0.57	0.40	0.30

**Table 5 molecules-26-00390-t005:** Particle size, polydispersity index (PDI), and zeta potential of optimized nanosuspensions stabilized in the mixture of Kollidon^®^ VA and Kollidon^®^ K12 along with one of Lutrol^®^ F127, hydroxypropyl β-cyclodextrin (HPβCD), polyethylene glycol (PEG) 6000 or sodium lauryl sulfate (SLS).

Formulations		Concentration (%, *w*/*v*)	Particle Size (nm) (Mean ± SD)	PDI (Mean ± SD)	Zeta Potential (mV)
F1	Kollidon^®^ VA/Kollidon^®^ K12/Lutrol^®^ F127	1.0/0.5/1.0	54.9 ± 1.8	0.29 ± 0.03	−35.2 ± 1.6
F2	Kollidon^®^ VA/Kollidon^®^ K12/HPβCD	1.0/0.5/1.0	43.0 ± 0.6	0.27 ± 0.01	−43.0 ± 2.3
F3	Kollidon^®^ VA/Kollidon^®^ K12/PEG 6000	1.0/0.5/1.0	53.1 ± 1.4	0.31 ± 0.02	−34.5 ± 1.8
F4	Kollidon^®^ VA/Kollidon^®^ K12/SLS	1.0/0.5/0.1	120.1 ± 2.2	0.33 ± 0.02	−40.1 ± 2.1

## Data Availability

The data underlying in this article will be shared on reasonable request to the corresponding author.

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
