# Peer review of "Solubility Determination of c-Met Inhibitor in Solvent Mixtures and Mathematical Modeling to Develop Nanosuspension Formulation"

_molecules, 2021, doi:10.3390/molecules26020390_

Round 1

Reviewer 1 Report

This paper deals with the determination of the solubility properties as well as the calculation of the solution thermodynamics of c-Met inhibitor, ABN401 in eleven mono-solvents and various (Transcutol-HP + water) mixtures at five different temperatures. The other hand, authors correlated the experimental data with some mathematical models. In general terms, the work is good, the data reported is of great importance and contributes to strengthening the solubility database of this therapeutic agent.
My remarks or objections are listed below:   
1. Abstract: The quantitative information is too low in the abstract. Authors are advised to include more quantitative information in order to enhance the readability of the manuscript.
2.            According to the guidelines of Molecules, the sections “results and discussion” should be section 2.

  1. The symbols in all equations should be italics and their subscripts or superscripts should be non-italics throughout the manuscript.
  2. In all equation "ln" must not be in italics
    5. The presentation of model/theoretical solubility with the subscript of “exp” is wrong because the subscript “exp” is used in the presentation of experimental solubility. Kindly change it throughout the manuscript.
  3. Dissolution studies: the method of dissolution studies for nanosuspension is wrong. The paddle method is used to study dissolution profile of solid dosage forms. Dialysis membranes are used to perform dissolution study from liquid formulations.
    7. Equation 6 is not appropriate to conclude what is established in page 12, last paragraph of section 3.3. I recommend performing the analysis using Wilson model.
    8. The sign of Gibbs energy is not an indicator of endothermic or exothermic processes (after equation (19) on page 13).
    9.            Describe the procedure for the preparation of co-solvent mixtures. was each mixture prepared only once? or was each mixture prepared in triplicate?.
    10. Kindly remove the list of figure captions from supplementary materials file.

Author Response

Reviewer #1:

This paper deals with the determination of the solubility properties as well as the calculation of the solution thermodynamics of c-Met inhibitor, ABN401 in eleven mono-solvents and various (Transcutol-HP + water) mixtures at five different temperatures. The other hand, authors correlated the experimental data with some mathematical models. In general terms, the work is good, the data reported is of great importance and contributes to strengthening the solubility database of this therapeutic agent.

The authors really appreciate the reviewer’s comments and recommendation. The reviewer’s input has been invaluable to the authors during the revision process. Responses to each comment are updated in the manuscript and highlighted in yellow.

My remarks or objections are listed below:

  1. Abstract: The quantitative information is too low in the abstract. Authors are advised to include more quantitative information in order to enhance the readability of the manuscript.

The authors appreciate the reviewer’s comment and recommendation. The authors tried to include more quantitative information within the 200-word limit. The authors also considered that the abstract needs to provide key information on the manuscript as well. The authors hope the reviewer consider the point.

  1. According to the guidelines of Molecules, the sections “results and discussion” should be section 2.

The authors really appreciate the reviewer’s comment. As suggested, the sections “results and discussion” are moved to section 2.

  1. The symbols in all equations should be italics and their subscripts or superscripts should be non-italics throughout the manuscript.

As pointed out, the symbols in all equations are thoroughly checked and subscripts or superscripts are corrected to non-italics in manuscript and supplementary file. The changes are highlighted.

  1. In all equation "ln" must not be in italics

As pointed out, the authors have corrected all equations with "ln" and highlighted the changes in yellow.

  1. The presentation of model/theoretical solubility with the subscript of “exp” is wrong because the subscript “exp” is used in the presentation of experimental solubility. Kindly change it throughout the manuscript.

The authors are so thankful for the reviewer’s keen point. The authors agree that the expression was inappropriate and may cause misunderstandings to the readers. The presentation in the model is corrected as “x1” with the subscript of “1”. The authors hope it avoid the confusion with the experimental mole fraction solubility.

  1. Dissolution studies: the method of dissolution studies for nanosuspension is wrong. The paddle method is used to study dissolution profile of solid dosage forms. Dialysis membranes are used to perform dissolution study from liquid formulations.

The authors appreciate the reviewer’s comment. The authors adopted the dissolution method for nanosuspension based on the reference “Development of a Resveratrol Nanosuspension Using the Antisolvent Precipitation Method without Solvent Removal, Based on a Quality by Design (QbD) Approach”, https://doi.org/10.3390/pharmaceutics. Actual particle size of the nanosuspension was bigger than around 40 nm and hence syringe filter can work for the sampling procedure. If dialysis is used, compatibility test needs to be performed to see any adsorption issue of the drug to the dialysis sac, which can be another diffusion barrier. As suggested by the reviewer, dialysis membranes need to be used if the formulation is liquid. However, for oral administration, simple dispersion or suspension would be good enough for the dissolution study.

  1. Equation 6 is not appropriate to conclude what is established in page 12, last paragraph of section 3.3. I recommend performing the analysis using Wilson model.

The authors appreciate the reviewer’s comment and recommendation. The authors revised the section 2.3 and updated the last sentence of the section as following: “The activity coefficient data support the favorable solubility in TWM mixture”. Moreover, please look at the equation 11 in Section 3.5. As recommended, the authors performed the analysis using the Wilson model and found that it might not be suitable for the manuscript based on the fitting results.

  1. The sign of Gibbs energy is not an indicator of endothermic or exothermic processes (after equation (19) on page 13).

The authors appreciate for reviewer’s keen point. Based on the reviewer’s comment, the sentence after the Equation 5 was revised to meet the comment.

  1. Describe the procedure for the preparation of co-solvent mixtures. was each mixture prepared only once? or was each mixture prepared in triplicate?.

The authors really appreciate the reviewer’s comment. The authors prepared the mixture in triplicate and its preparation of co-solvent mixtures was consistent with the various solvents. The changes are updated in the last paragraph of section 3.4.

  1. Kindly remove the list of figure captions from supplementary materials file.

As asked by the reviewer, the list of figure captions are removed from the supplementary materials file.

Reviewer 2 Report

The authors studied “Solubility determination of c-Met inhibitor in solvent mixtures and mathematical modelling to develop nanosuspension formulation”. The minor concern needs to be addressed before the paper is accepted for publication. Followings are recommended for the manuscript. 

What is the novelty of work?

The quality of the figures is poor

The vocabularies, grammar and writing style in the manuscript must be revised and improved.

Authors should define abbreviation first time after that can use abbreviation.

Parameters optimized while formulating nanosuspension need to be presented in tabular form with particle size, PDI and zeta potential as shown in this reference “Development of Polymer and Surfactant Based Naringenin Nanosuspension for Improvement of Stability, Antioxidant, and Antitumour Activity”. https://doi.org/10.1155/2020/3489393

The author did not mention the freeze-drying procedure and cryoprotectant and its percentage used.

How authors have managed with sink condition for in vitro dissolution of nanosuspension as the drug is poorly soluble.

Stability data can be incorporated if performed.

Author Response

Reviewer #2:

The authors studied “Solubility determination of c-Met inhibitor in solvent mixtures and mathematical modelling to develop nanosuspension formulation”. The minor concern needs to be addressed before the paper is accepted for publication.

The authors really appreciate the reviewer’s comments and recommendation. The reviewer’s input has been invaluable to the authors during the revision process. Responses to each comment are updated in the manuscript and highlighted in yellow.

Followings are recommended for the manuscript. What is the novelty of work?

The authors really appreciate the reviewer’s comment. Since it is a new drug candidate, its solubility data were needed for the purposes of analysis and drug development. Based on the solubility results, formulation strategies would be selected including liquid (nano/micro-suspension or solution in oils/surfactants) and solid formulations. Especially, liquid formulations would be suitable for rodents due to the challenges for administration. In this study, various solubility data were obtained with systematic and reasonable approaching method to gain data about the most suitable solvent systems for the drug candidate. It will be very useful when to deal with new chemical entities for developing liquid formulation such as nanosuspension especially using liquid anti-solvent precipitation method. The authors hope the reviewer consider this point.

The quality of the figures is poor

The quality of the figures was improved to 1000 dpi and added in the manuscript. Especially, quality of Figure 2 increased even more to 1200 dpi. The authors can provide the original figures if necessary, during the editing process.

The vocabularies, grammar and writing style in the manuscript must be revised and improved.

The authors appreciate the reviewer’s comment. The manuscript was already gone through the English editing service with a native speaker. However, as asked, the manuscript was carefully revised to improve grammar and vocabularies.

Authors should define abbreviation first time after that can use abbreviation.

The authors appreciate the reviewer’s comment. The authors have gone through the manuscript and ensure the abbreviation is defined at the first mention.

Parameters optimized while formulating nanosuspension need to be presented in tabular form with particle size, PDI and zeta potential as shown in this reference “Development of Polymer and Surfactant Based Naringenin Nanosuspension for Improvement of Stability, Antioxidant, and Antitumour Activity”. https://doi.org/10.1155/2020/3489393

The authors appreciate the reviewer’s comment. As suggested, the authors have presented the information of particle size, PDI and zeta potential in tabular form (Table 5).

The author did not mention the freeze-drying procedure and cryoprotectant and its percentage used.

The authors really appreciate the reviewer’s comment. As suggested by the reviewer, the authors have briefly mentioned the freeze-drying procedure and used cryoprotectant along with its percentage used. The authors can tell that the reviewer is interested in the freeze-drying process starting from freezing, primary drying, and secondary drying together with lyo-protectant. The authors wanted to obtain the dried samples to see the appearance and did not pay attention on the process development. If so, FDM (freeze drying microscope) and DSC would be performed to get critical temperatures, which might beyond the scope of current study.

How authors have managed with sink condition for in vitro dissolution of nanosuspension as the drug is poorly soluble.

The authors really appreciate the reviewer’s point. The authors adopted dissolution method from reference “New Preclinical Development of a c-Met Inhibitor and Its Combined Anti-Tumor Effect in c-Met-Amplified NSCLC”, https://doi.org/10.3390/pharmaceutics12020121. The authors estimated sink condition values based on calculation (sink condition = saturation solubility of ABN401 in 500 mL medium (pH 1.2)/ABN401 dose). The equivalent amount of aliquot was replaced with fresh medium in the dissolution vessel each time to ensure the sink condition was maintained throughout the experiment. The authors hope it work for the reviewer.

Stability data can be incorporated if performed.

The authors really appreciate the reviewer’s comment. As asked, the authors included the stability data of 3 days and 7 days with respect to particle size and zeta potential in Table S8. The authors hope it may work for the reviewer.

Round 2

Reviewer 1 Report

Authors have addressed the previous concerns. Revised manuscript can be accepted for the publication in its present form.